# Cannabis Dopaminergic Effects Induce Hallucinations in a Patient with Parkinson’s Disease

**DOI:** 10.3390/medicina57101107

**Published:** 2021-10-14

**Authors:** Katie Pizzolato, David Thacker, Nicole Del Toro-Pagán, Abeer Hanna, Jacques Turgeon, Adriana Matos, Nishita Amin, Veronique Michaud

**Affiliations:** 1Office of Translational Research and Residency Programs, Tabula Rasa HealthCare, Moorestown, NJ 08057, USA; kpizzolato@trhc.com (K.P.); npagan@trhc.com (N.D.T.-P.); amatos@carekinesis.com (A.M.); namin@carekinesis.com (N.A.); 2Tabula Rasa HealthCare, Precision Pharmacotherapy Research and Development Institute, Tabula Rasa HealthCare, Orlando, FL 32827, USA; dthacker@trhc.com (D.T.); jturgeon@trhc.com (J.T.); 3Viecare Butler, Program of All-Inclusive Care for the Elderly (PACE), Butler, PA 16001, USA; abeer.hanna@lutheranseniorlife.org; 4Faculty of Pharmacy, Université de Montréal, Montréal, QC H3C 3J7, Canada

**Keywords:** cannabis, Parkinson’s disease, pharmacogenomics, COMT, CYP2C9, hallucinations

## Abstract

Cannabis products that contain the tetrahydrocannabinol (THC) cannabinoid are emerging as promising therapeutic agents for the treatment of medical conditions such as chronic pain. THC elicits psychoactive effects through modulation of dopaminergic neurons, thereby altering levels of dopamine in the brain. This case report highlights the complexity associated with medicinal cannabis and the health risks associated with its use. A 57-year-old male with Parkinson’s disease was experiencing worsening tremors and vivid hallucinations despite therapy optimization attempts. It was discovered that the patient took cannabis for chronic back pain, and a pharmacogenomics (PGx) test indicated the presence of variants for the *COMT* and *HTR2A* genes. These variants could increase dopamine levels and predispose patients to visual hallucinations. Once the cannabis was discontinued, the patient’s hallucinations began to slowly dissipate. Cannabis use continues to expand as it gains more acceptance legally and medicinally, but cannabis can affect the response to drugs. This patient case suggests that cannabis use in combination with dopamine-promoting drugs, especially in a patient with genetic variants, can increase the risk for vivid hallucinations. These conditions support the importance of considering herb–drug interactions and PGx data when performing a medication safety review.

## 1. Introduction

The use of medical cannabis has steadily gained popularity over the last several years. Cannabis has demonstrated therapeutic effects with different cannabinoids derived from the cannabis plant, specifically tetrahydrocannabinol (THC) and cannabidiol (CBD), which are being utilized for several medical conditions by providing analgesic, antispasmodic, and antiemetic properties [1,2,3]. While the role of cannabis in medicine continues to expand, it is imperative to consider cannabis effects and potential drug interactions. Research has demonstrated that THC and CBD are substrates of the cytochrome P450 (CYP) enzymes CYP2C9 and CYP2C19, and thus, will increase the risk for drug–drug interactions [4]. Additionally, THC has been shown to elicit its psychoactive effects through modulation of dopaminergic neurons, thereby altering levels of dopamine in different areas of the brain [5].

Appropriate dopamine levels are vital, as the nervous system utilizes dopamine for the regulation of several physiological functions (e.g., mood, motor, cognitive) [6]. Enzymes and transporters, such as the catechol-O-methyltransferase (COMT) enzyme and dopamine transporters, help regulate the level of dopamine in the synapses [6]. The *COMT* gene is responsible for producing the COMT enzyme, which acts as a metabolizing enzyme for dopamine [6]. Therefore, genetic alterations of *COMT* and other relevant genes affecting dopamine can potentially alter how an individual responds to THC [7].

Low levels of dopamine in specific brain regions have been found to be associated with certain conditions, such as Parkinson’s disease (PD). PD is a neurodegenerative disease caused by the death of dopamine-producing neurons in the substantia nigra, which is a part of the midbrain responsible for coordinating movement [8]. Neuronal death in this region results in low dopamine concentrations; therefore, treatment strategies for PD often involve initiating medications (e.g., carbidopa-levodopa, ropinirole, entacapone) that result in increased concentrations of dopamine to improve movement control [8]. Concomitant intake of cannabis and drugs used in the treatment of PD can significantly modulate dopamine concentrations. The case described herein demonstrates the importance of considering the pharmacokinetic and pharmacodynamic effects of cannabis and an individual’s pharmacogenomic (PGx) data when evaluating a patient’s medication regimen for therapeutic response and/or adverse drug events.

## 2. Case Presentation

A 57-year-old male with a past diagnosis of PD began treatment with a new primary care physician (PCP). In addition to PD, the patient’s past medical history includes spinal stenosis, vitamin D deficiency, frequent falls, a history of nicotine and alcohol dependence, mild kidney disease, and chronic neck, back, and shoulder pain. The patient had spinal surgery approximately two years ago, during which he developed complications (i.e., delirium) from anesthesia, causing him to remain hospitalized for an additional month. During his initial meeting with the PCP, the patient reported experiencing worsening tremors, body pain, and vivid visual hallucinations encompassing small children and flying objects, which the patient claimed to have been seeing for over two years. This patient was also prescribed rivastigmine, as Lewy body dementia was recently included as a differential diagnosis due to the presence of these hallucinations. Upon evaluation of the patient’s past medical history and medications, changes were made to his drug regimen to better control his symptoms, which included increasing the doses for carbidopa-levodopa and rivastigmine (Figure 1). The patient’s tremors began to slowly diminish with his new medication regimen; however, his vivid hallucinations were still present. Upon further inquisition at a subsequent visit, the PCP discovered that approximately two years ago the patient had been advised to chronically use cannabis to manage his chronic back pain. On average, the patient reported smoking approximately 3 g of cannabis per week. The new PCP promptly recommended the cessation of cannabis, as its use with other medications could be contributing to his vivid hallucinations. The patient’s interdisciplinary team continued to evaluate his hallucinations and tremors after discontinuation of cannabis, which was confirmed with a negative drug screen. The patient reported that his hallucinations began to diminish slowly over time. As more time elapsed, his hallucinations of children and flying objects changed to seeing only floating dots. Considering the improvement in hallucinations was observed following discontinuation of cannabis, rivastigmine was discontinued as it was determined that the patient’s hallucinations were likely not due to Lewy body dementia. During a follow-up encounter two months later, the patient reported that his hallucinations had disappeared, and control of his tremors had improved further. The clinical pharmacist recommended dose increases of carbidopa-levodopa to help control his tremors still present early in the morning (Figure 1).

A PGx test was also ordered to help determine if a genetic component could explain why this patient experienced such vivid hallucinations from cannabis. Upon reception of the PGx results (Table 1), the pharmacist observed that the patient was homozygous for the *COMT* variant (rs4680 AA (Met/Met); this variant is associated with low enzyme (COMT) activity due to a decreased production of the enzyme. Such a decrease in COMT activity is associated with higher levels of catecholamines (i.e., dopamine) in the brain.

To manage his chronic pain (after cessation of cannabis), the patient was prescribed celecoxib and lidocaine topical patches for back, neck, and shoulder pain. However, his PCP determined that additional medication for pain control was warranted. Given that patient was identified as a CYP2D6 intermediate metabolizer (Table 1; *CYP2D6*1/*4*), the clinical pharmacist recommended prescribing an opioid that does not utilize the CYP2D6 pathway (e.g., morphine) to decrease the risk of pharmacotherapy failure and/or possible adverse drug events.

## 3. Discussion

The use of cannabis products for medical purposes continues to expand as research develops. The patient under consideration in this case report was initially recommended cannabis for analgesia due to chronic back pain. Guidance is currently available regarding medical cannabis use for the treatment of chronic pain, suggesting that cannabis-based drugs can be considered when all other treatment options have failed [1]. CBD and THC are the two most prominent cannabinoids found in cannabis and have been used to treat multiple sclerosis spasms, neuropathic and cancer pain, nausea, and insomnia [4]. CBD has been utilized as an anxiolytic and the U.S. Food and Drug Administration has approved the CBD oral solution (Epidiolex^®^) as a treatment option for epilepsy [4,11]. While research has provided guidance on managing chronic pain with cannabis, guidance is not available regarding use in other conditions. However, research has shown that cannabis use can cause hallucinations in patients with PD, thus the appropriateness of cannabis use should be evaluated [1,12].

THC is the main psychoactive cannabinoid in cannabis as both the parent molecule and its 11-hydroxy-delta-9-tetrahydrocannabinol (11-OH-THC) metabolite produce euphoric effects [13]. THC does not provide analgesic effects, but its 11-carboxy-delta-9-tetrahydrocannabinol (11-COOH-THC, psychotropically inactive) metabolite possesses anti-inflammatory and analgesic properties [13]. The non-psychoactive analogue of THC, CBD, another cannabinoid found in cannabis, has also shown analgesic and anti-inflammatory effects [14].

THC is metabolized by CYP enzymes in the liver, particularly CYP3A4 and CYP2C9 [4,15]. CBD is mainly metabolized by CYP3A4 and CYP2C19, and at higher oral doses (5 mg/kg/day), can inhibit CYP2C9 and to a lesser extent CYP1A2 [8,11,16]. Concomitant administration of prescribed medications with cannabis engenders a risk of potential herb–drug CYP450 interactions (Figure 2). Therefore, any drug with a stronger affinity for the CYP2C9, CYP2C19 or CYP3A4 enzymes than THC or CBD, if administered at the same time, could affect their disposition and result in an herb–drug interaction [4,17]. These interactions could lead to increased CBD or THC concentrations and possibly lower concentrations of THC’s metabolites [4,17]. Furthermore, CYP2C9 and CYP2C19 are both polymorphic enzymes with differing functions that could modulate cannabis exposure if genetic variants are present (which was excluded as a contributing factor for this case) (Table 1) [15,18].

Although studies investigating the use of cannabis for pain demonstrated mixed results, there is emerging evidence supporting the benefit of cannabis for pain [19,20,21]. Numerous factors can explain discrepancies between study results such as pain models, healthy subjects vs. patients, routes of administration (inhalation vs. oral), and sources of the product [19,20,21]. The utility of cannabis use remains under debate as there is no approved indication, formulation, or dosage for pain. Further research is needed to better understand the efficacy, dose–response effects, routes of administration and side-effect or safety profiles associated with such products. In general, inhaled cannabis (smoking and vaping) is associated with a quicker onset of action, while oral administration of cannabis has a slower or delayed onset of action, and it is exposed to the intestinal-hepatic first-pass effect [4]. Side effects and safety profiles should be considered for both routes of administrations.

PD is characterized by the death of dopamine-producing neurons, specifically in the substantia nigra, which impacts an individual’s ability to control their movements [8]. Treatment strategies to combat PD motor symptoms include drugs such as dopamine precursors (e.g., carbidopa-levodopa), dopamine agonists (e.g., ropinirole), COMT inhibitors (e.g., entacapone), and monoamine oxidase B inhibitors (e.g., rasagiline) [8]. Additionally, visual hallucinations are a common non-motor symptom observed in patients with PD, and they typically result from long-term use or dose increases of PD drugs [8]. The cause of these particular hallucinations can be multifactorial but there is evidence attributing them to high levels of dopamine [22]. In this patient’s case, the medical team considered the patient’s PD medication regimen and disease progression when attempting to identify the cause of hallucinations while managing worsening tremors. The clinical presentation suggested that the patient was not receiving enough dopamine from his medications to control his tremors supporting the proposed increased dose of carbidopa-levodopa. In addition, the dose of rivastigmine was increased to improve cognitive and functional abilities and diminish visual hallucinations [23]. These pharmacotherapy interventions improved his tremors; however, his vivid hallucinations were not alleviated which warranted further consideration.

The endocannabinoid system (ECS) is an essential regulator of dopamine levels [4,24]. The ECS is a neuromodulatory system able to regulate several neurons (e.g., dopamine) [4,24,25]. THC acts as a receptor agonist for receptors in the ECS, known as cannabinoid receptor 1 (CB1R) and cannabinoid receptor 2 (CB2R) [4,5,24]. CB1Rs are found predominately in the brain and are located on many neurons presynaptically, as well as postsynaptically [26]. In regards to dopamine neurons, inhibitory neurotransmitters such as gamma-aminobutyric acid (GABA) act on dopamine neurons to regulate and reduce dopamine release into the synapses [5,24,25]. When CB1Rs on presynaptic neurons are activated, inhibitory neurotransmitter levels are reduced and dopamine levels increase [5,24,25]. When THC binds in the striatum and cortex, the euphoric feeling associated with THC occurs; however, abnormal levels of dopamine in the striatal and limbic regions of the brain have been observed in patients experiencing psychotic symptoms, including hallucinations [5,24,27].

A small percentage of people have been shown to experience psychotomimetic effects with low-dose THC in the presence of genetic polymorphisms [7]. The *COMT* gene is responsible for producing the COMT enzyme, which acts as a metabolizing enzyme for dopamine [6]. The COMT enzyme demonstrates the importance of dopamine regulation within the prefrontal cortex, as there are fewer available dopamine transporters in this region of the brain [28]. In *COMT*, a common polymorphism can occur due to a change from the amino acid valine (Val) to a methionine (Met) [Val158Met], which results in a reduction in dopamine metabolism [29]. Therefore, carriers of the Met variant experience higher dopamine concentrations in synapses [29]. Studies in healthy individuals with the *COMT* wild-type have been shown to metabolize dopamine up to four times faster than those with *COMT* Met/Met, deeming individuals homozygous for Met as having low COMT enzyme activity [28]. Interestingly, the patient’s PGx results (Table 1) reported a genotype for *COMT* Met/Met, therefore higher levels of dopamine in his synapses would be expected.

Research has been conducted to evaluate the effects of the *COMT* Val158Met polymorphism and the use of cannabis on the risk for psychosis [28,30]. In healthy individuals, there is an observed correlation between cannabis use in those with *COMT* Val/Val and increased risk for psychosis [28,30]. This genotype is also associated with lower levels of prefrontal dopamine; therefore, this observation is thought to be due to higher levels of phasic dopamine transmission, which aids in the development of psychotic symptoms [28]. Additionally, a previous study evaluated patients with PD and their subsequent reactions to cannabis use, reporting that one out of five patients experienced hallucinations [12]. Given these studies and considering THC can also increase dopamine levels, extra care should be exercised in carriers of the Met variant in regards to cannabis use [31].

Furthermore, this patient was reported to have a single nucleotide polymorphism on the *HTR2A* gene, resulting in altered serotonin 2A (5-HT2A) receptor function (Table 1). While there is a lack of clinical studies with the 5-HT2A receptor and cannabis, research has demonstrated that the CB1R is expressed on serotonin neurons; therefore, the binding of THC to this receptor increases neuronal firing [32]. In pre-clinical studies, the CB1R and 5-HT2A have been shown to form a heteromer that could be responsible for altered homeostasis of the serotonin system [33]. This type of serotonin receptor is commonly found in the prefrontal cortex and evidence has suggested that alterations in receptor function have demonstrated association with mood disorders and psychosis [25]. Several studies have proven that cannabinoids affect the serotonergic system; however, the data regarding the relationship between cannabis and *HTR2A* and its subsequent effects are limited [25].

The PGx test results identified the patient as a CYP2D6 intermediate metabolizer, which is heavily involved in the metabolism of several opioid drugs utilized for pain (e.g., codeine, tramadol, hydrocodone, oxycodone) [34]. Guidelines are available for *CYP2D6* and select opioids, which state that individuals with CYP2D6 intermediate metabolizer status may experience altered effects (e.g., possible adverse drug events, increased risk for pharmacotherapy failure) when compared to CYP2D6 normal metabolizers [34]. Given this patient’s genetic results, guidelines, and current pain status, the clinical pharmacist suggested morphine as an alternative, as it does not utilize CYP2D6 for metabolism [34]. Studies have been conducted evaluating pain sensitivity in individuals with *COMT* genotype Val/Met and Met/Met, reporting that those with this polymorphism have an increased pain sensitivity [35]. In addition, low COMT activity has demonstrated association with increased opioid analgesia and opioid side effects (e.g., nausea, vomiting) [35]. Considering these studies and the patient’s *COMT* genotype, if an opioid were to be prescribed, starting a low dose opioid and monitoring for any medication-related side effects would be suggested. There are additional genetic polymorphisms proven to affect an individual’s reaction to THC, such as the *Taq1A* polymorphism in the dopamine receptor gene; however, these data were not available for the patient at the time of these interventions. [7].

## 4. Conclusions

Cannabis use has been demonstrated to have an impact on dopamine concentrations in the brain, resulting in side effects like hallucinations. In a condition like PD, many interactions and their subsequent side effects can occur when combining cannabis with dopamine-promoting drugs and the genetic variants that affect dopamine. This patient case demonstrates the importance of considering cannabis use when evaluating for potential drug–drug and drug–gene interactions in an individual’s regimen. Considering all drug-use (e.g., prescribed, recreational, over-the-counter drugs), along with an individual’s PGx results when evaluating a patient allows for a safer and more accurate approach when completing comprehensive assessments.

## Figures and Tables

**Figure 1 medicina-57-01107-f001:**
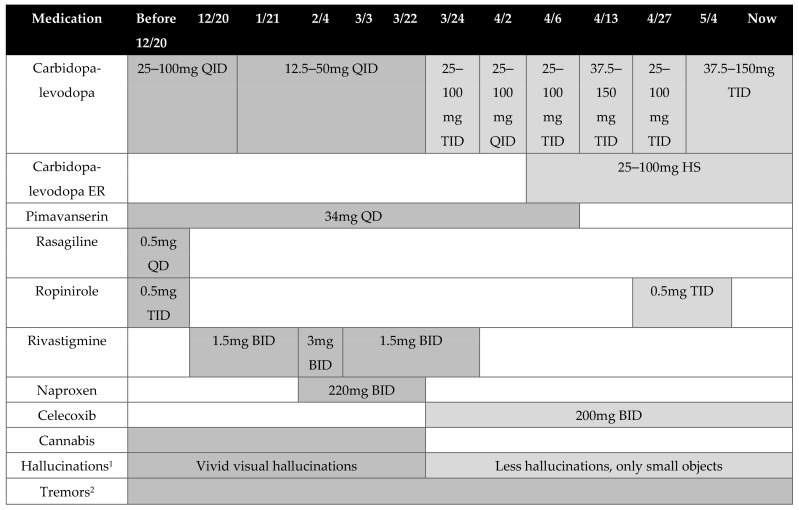
Drug and Symptom Change Timeline. ^1^ Shaded area in the chart represents when hallucinations were experienced by the patient. Shades of gray are associated with the presence of vivid or fewer hallucinations and which medications are administered at that time. ^2^ Period when tremor symptoms were experienced by the patient. Abbreviations: QD: once a day, BID: twice a day, TID: three times a day, QID: four times a day, HS: at bedtime.

**Figure 2 medicina-57-01107-f002:**
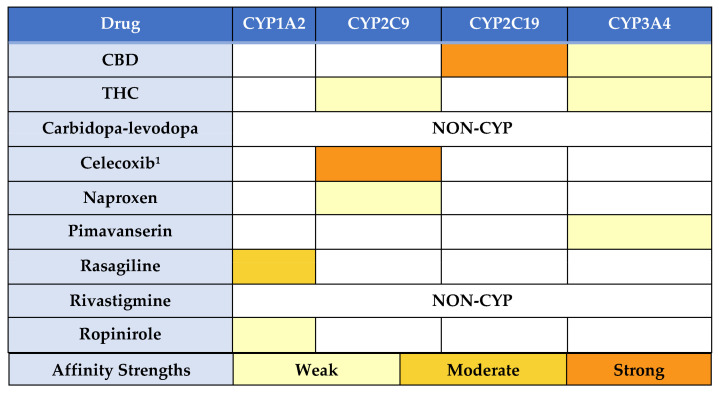
Summary of Affinity and CYP Metabolic Pathways. ^1^ Treatment with celecoxib began after the cessation of cannabis. Abbreviations: CBD: cannabidiol, CYP: Cytochrome P450, THC: tetrahydrocannabinol.

**Table 1 medicina-57-01107-t001:** Patient’s Pharmacogenomics Test Results.

Gene	Genotype	Phenotype
*CYP1A2* ^1^	**1F|*1F*	Possible Normal Metabolizer
*CYP2B6*	**1|*1*	Normal Metabolizer
*CYP2C9*	**1|*1*	Normal Metabolizer
*CYP2C19*	**1|*1*	Normal Metabolizer
*CYP2D6*	**1|*4*	Intermediate Metabolizer
*CYP3A4* ^2^	**1|*1*	Undetermined
*CYP3A5*	**3|*3*	Poor Expresser Metabolizer
*COMT*	*rs4680 AA (Met/Met)*	Low Activity
*DRD2*	*rs1799978 AA*	Normal Receptor Expression
*HTR2A*	*rs7997012 GG*	Altered Receptor Function

^1^ Common variants in *CYP1A2* gene reflect its inducibility. *CYP1A2* genetic variations, without the presence of induction (e.g., smoking, concomitant CYP1A2 inducers), have not been demonstrated to clinically alter the activity of CYP1A2 [9]. ^2^ CYP3A4 gene shows some genetic variations and most variants have not been demonstrated to clinically alter the activity of CYP3A4. Many of the variants are extremely rare, making it difficult to derive a phenotype based on genetic results [10].

## Data Availability

Not applicable.

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
