# Peer review of "Cannabis Dopaminergic Effects Induce Hallucinations in a Patient with Parkinson’s Disease"

_medicina, 2021, doi:10.3390/medicina57101107_

Round 1
Reviewer 1 Report
Dear authors,
I would like to congratulate You for Your work. There are, however, a few points that need modifications and clarifications
- please adjust the author names and affiliations to correspond with the journals format
- Line 52 ``Low levels of dopamine in specific brain regions have been found to cause certain conditions, such as Parkinson’s disease (PD)`` - please clarify since, it is stated that PD associates with decrease in dopamine levels like You specified, however as a consequence of PD not as a ethyological factor of PD.
- In Figure 1, i don`t understand why Cannabis is mentioned if it is not involved and also the tremors are not characterised, please make the Figure more clear for readers and add the missing information
- Also, regarding the patients history, line 66 the patient’s past medical history includes Lewy Body dementia - a condition which is accompanied by hallucinations, often in the first few years - please clarify what was the evolution of this diagnosis and how did the authors exclude it as a cause for the vivid hallucinations?
- a spell check would also be advised
Author Response
The authors would like to thank both of the reviewers for their time and comments put towards our case report. We have worked to address each comment raised and hope these explanations and/or modifications offer clarification for each point.
Reviewer 1:
- I would like to congratulate You for Your work.
We thank the reviewer for taking time to review the manuscript and appreciate positive comments.
- Please adjust the authors names and affiliations to correspond with the journals format.
The authors names and affiliations were corrected according to the journal format.
- Line 52 ``Low levels of dopamine in specific brain regions have been found to cause certain conditions, such as Parkinson’s disease (PD)`` - please clarify since, it is stated that PD associates with decrease in dopamine levels like You specified, however as a consequence of PD not as a ethyological factor of PD.
We understand the reviewer’s point made here. To address the comment, the sentence was modified as follows (page 2 in the Introduction section, Line 53):
“Low levels of dopamine in specific brain regions have been found to be associated with certain conditions, such as Parkinson’s disease”.
- In Figure 1, I don’t understand why Cannabis is mentioned if it is not involved and also the tremors are not characterized, please make the Figure more clear for readers and add the missing information
Figure 1 was enlarged and gray shades were used to help the visualization. Cannabis was mentioned because it could be involved with the presence of hallucinations; therefore, we felt it was important to include the patient’s use of cannabis to be able to compare this to the changes in the patient’s hallucinations. In regard to the characterization of the tremors, the severity of the patient’s tremors fluctuated often with changes in the patient’s medications, sometimes changing between a few days or between specific times within a single day. This can be seen in Figure 1 as the carbidopa-levodopa dose was changed multiple times in a span of a few weeks. Characterizing the severity of the tremors and when they occurred in the chart would be difficult to portray in Figure 1, as there is not necessarily a concise way to specify these changes. Ultimately the goal was complete management of tremor symptoms, which was not achieved during the timeframe discussed in Figure 1.
A footnote was added as follows:
1Shaded area in chart represents when hallucinations were experienced by the patient. Shades of gray are associated with the presence of vivid or less hallucinations and which medications are administered at that time.
2Period when tremor symptoms were experienced by the patient.
- Also, regarding the patients history, line 66 the patient’s past medical history includes Lewy Body dementia - a condition which is accompanied by hallucinations, often in the first few years - please clarify what was the evolution of this diagnosis and how did the authors exclude it as a cause for the vivid hallucinations?
The reviewer raised a great point. Lewy Body dementia (LBD) was included in the differential diagnosis later on in the patient’s treatment as a potential explanation for the patient’s hallucinations. Once it was found that the medication (rivastigmine) was not helping the patient’s hallucinations, and that the patient used cannabis, rivastigmine was discontinued. It was determined by the patient’s physician that his hallucinations were not from the diagnosis of LBD and was justified by the cessation of cannabis resulting in a decrease in hallucinations.
To address this comment in this case report, a sentence was added (section 2 in the Case presentation, Line 77):
“This patient was also prescribed rivastigmine, as Lewy Body dementia was recently included as a differential diagnosis due to the presence of these hallucinations.”
Additionally, to better understand the sequence of the medical history/diagnosis, “Lewy Body dementia” was removed on page 2 Line 68, and mentioned in Line 75.
This sentence was added (page 2 in the Case Presentation section, Line 90):
“Considering improvement in hallucinations was observed following discontinuation of cannabis, rivastigmine was discontinued as it was determined that the patient’s hallucinations were likely not due to Lewy Body dementia.”
- a spell check would also be advised
This report has been reviewed again by a native English speaker.
Reviewer 2 Report
The authors demonstrated the interesting case on Cannabis use in PD for pain control. Need to clarify the point regarding some conflicts over medicinal use of cannabis. In this case the patient reported smoking 'approximately 3 grams' of cannabis per week not the oral form of substances. Need more details about the discussion on pos and cons in smoking cannabis especially regarding the dose and the feasibilty in medical aspect.
Author Response
The authors would like to thank both of the reviewers for their time and comments put towards our case report. We have worked to address each comment raised and hope these explanations and/or modifications offer clarification for each point.
1. The authors demonstrated the interesting case on Cannabis use in PD for pain control. Need to clarify the point regarding some conflicts over medicinal use of cannabis. In this case the patient reported smoking 'approximately 3 grams' of cannabis per week not the oral form of substances. Need more details about the discussion on pos and cons in smoking cannabis especially regarding the dose and the feasibility in medical aspect
The authors thank this reviewer for taking the time to review our case report. It was found that the cannabis was recommended by a physician outside of the PACE center for the patient’s chronic back pain. The pros and cons of medicinal use of cannabis are good points. The authors feel that this could be out of scope of this case, as we aimed to analyze cannabis and its effects on hallucinations. Additional comments were added to briefly address this point as it was described in details in previous publications. Additional references on this specific topic are now provided. In the Discussion section (Line 166):
“Although studies investigating the use of cannabis for pain demonstrated mixed results, there is emerging evidence supporting the benefit from cannabis for pain [17-19]. Numerous factors can explain discrepancies between study results such as pain models, healthy vs patients, routes of administration (inhalation vs oral), and sources of the product [17-19]. The utility of cannabis use remains under debate as there is no approved indication, formulation, or dosage for pain. Further research is needed to better understand the efficacy, dose-response effects, routes of administration and side-effect or safety profiles associated with such products. In general, inhaled cannabis (smoking and vaping) is associated with quicker onset of action, while oral administration of cannabis has a slower or delayed onset of action and it is exposed to the intestinal-hepatic first pass effect [4]. Side effects and safety profiles should be considered for both routes of administrations.”
Round 2
Reviewer 1 Report
Thank you very much for the modifications and clarifications, i have no further comments